# The Role of Climatic, Environmental and Socioeconomic Factors in the Natural Movement of Urban Populations in Kazakhstan, 2012–2020: An Analysis from a Middle-Income Country in Central Asia

**DOI:** 10.3390/ijerph21040416

**Published:** 2024-03-29

**Authors:** Nurlan Smagulov, Olzhas Zhamantayev, Aiman Konkabayeva, Ainur Adilbekova, Gulmira Zhanalina, Nurzhamal Shintayeva, Dinara Aubakirova

**Affiliations:** 1Research Park of Biotechnology and Eco-Monitoring, Karaganda Buketov University, Karaganda 100028, Kazakhstan; 2School of Public Health, Karaganda Medical University, Karaganda 100000, Kazakhstan; 3Faculty of Biology and Geography, Karaganda Buketov University, Karaganda 100028, Kazakhstan; 4Department of Morphology and Physiology, Karaganda Medical University, Karaganda 100000, Kazakhstan; adilbekova@qmu.kz

**Keywords:** demography, urban population, socioeconomic factors, climate, environment, public health, Kazakhstan, central Asia, middle-income country

## Abstract

Background: This study addresses the importance of identifying key characteristics influencing demographic indicators for urban populations, emphasizing the need to consider regional climatic features and ecological factors. The research utilized data from ten main regional cities across the Republic of Kazakhstan. Methods: This study involved a retrospective analysis based on secondary data from official sources spanning 2012–2020. We employed correlation analysis and multidimensional regression models. Results: Noteworthy predictors for crude birth rate included the influence of effective temperature (β = 0.842, *p* < 0.0001), marriage rate (β = 0.780, *p* < 0.0001), Gini coefficient (β = −27.342, *p* = 0.020) and divorce rate (β = −2.060, *p* < 0.0001), with overall strong model performance (R^2^ = 0.940). The degree of atmospheric pollution (β = −0.949, *p* = 0.044), effective temperature (β = −0.294, *p* < 0.0001) and Gini coefficient (β = 19.923, *p* = 0.015) were the predictors for crude mortality rate, with a high model fit (R^2^ = 0.796). Conclusions: The study unveils significant relationships between demographic indicators (crude birth rate, mortality rate) and variables like effective temperature, marriage rate, divorce rate, Gini coefficient, physician density and others. This analysis of climatic, environmental, and socioeconomic factors influencing demographic indicators may help in promoting specific measures to address public health issues in Kazakhstan.

## 1. Introduction

The demographic projections of the United Nations for 2022 state that the world’s demographic landscape in the 21st century will be shaped by ongoing trends, including a decline in fertility rates, an increase in life expectancy, demographic transitions, migration, and other factors [1].

Demographic indicators are crucial to the health status of populations, acknowledging that climatic conditions, human-induced environmental pollution and the lifestyle choices of the population play significant roles [2,3]. Kazakhstan presents an interesting case study due to its industrial nature. Dominated by mining, metallurgical, chemical, and energy enterprises, these aspects present notable ecological challenges that impact not only the local ecology but also public health. Recognizing this confluence of factors, our study divides them into three broad categories: climatic, environmental, and socioeconomic. It emphasizes the need for comprehensive analysis methods, estimates, and predictive models to understand how these environmental phenomena affect population dynamics [4]. The primary goal is to enhance population quality of life and demographic dynamics, necessitating measures to mitigate the adverse effects of climatic and environmental conditions while improving socioeconomic indicators. WHO reports indicate that 1 to 10% of annual deaths in Europe are attributed to adverse climatic and weather factors [5]. Sociodemographic factors such as socioeconomic deprivation, education level, income, marital status, and professional class serve as distorting factors or effect modifiers, exerting a notable influence on demographic indicators [6,7,8,9,10].

Meanwhile, healthcare system efficiency plays an important role in enhancing life expectancy; several studies even suggest a correlation between higher GDPs per capita and increased life expectancy [11,12,13,14]. This relationship suggests that wealthier nations tend to have better healthcare outcomes, likely due to more resources being invested into health infrastructure, technologies, and services. Furthermore, physician density and nurse density—key measures of healthcare accessibility—also contribute significantly to demographic indicators. Countries with a higher density of medical professionals tend to have lower mortality rates and higher life expectancies [15,16]. This could be because a greater availability of doctors and nurses allows for easier access to preventative care and early disease detection, leading to improved health outcomes. In densely populated areas where healthcare resources may be stretched thin, having an adequate number of medical professionals can make a significant difference in managing population health. Likewise, in less populated or rural regions, sufficient physician and nurse density ensures that individuals receive necessary care without enduring long travel distances.

Understanding the processes influencing natural population movement indicators is necessary for enhancing planning, organization, resource allocation, and potential intervention strategies. Developing effective mathematical forecasting models requires the identification of appropriate predictors, emphasizing the need to discern factors determining changes in demographic indicators. This comprehensive approach enables the consideration of regional climatic features, ecological aspects, and other relevant factors. The aim of our study was to conduct a complex assessment, evaluating the influence of climatological, environmental, and socioeconomic factors on the dynamics of natural population movement in the Republic of Kazakhstan.

## 2. Materials and Methods

Information was collected from 10 main cities in all regions of Kazakhstan, including Aktobe, Atyrau, Karaganda, Kostanay, Kyzylorda, Pavlodar, Petropavlovsk, Temirtau, Uralsk and Ust-Kamenogorsk. Retrospective studies involved the utilization of secondary data obtained from official sources, namely, the Bureau of National Statistics, the Ministry of Ecology, Geology and Natural Resources of the Republic of Kazakhstan and the Weather and Climate website spanning the years 2012 to 2020.

Weather conditions were assessed using the “Effective temperature” (ET), a bioclimatic indicator that integrates both temperature and humidity effects into a single measure that can better represent human comfort level in varying weather conditions than average temperature or humidity alone. Its units are degrees Celsius (°C). Data for this indicator were sourced from the Weather and Climate website, http://www.pogodaiklimat.ru (accessed on 6 December 2023) (Figure 1). The ET serves as an index of weather comfort that is widely employed in numerous countries.

The state of atmospheric air pollution was evaluated using the “Standard Index” (SI), derived from the monthly bulletins of Kazhydromet [17]. The SI categorizes atmospheric pollution into four standard grades: 1—low, 2—elevated, 3—high, and 4—very high.

Quantitative data for various variables, such as “Gross Regional Product” (GRP per capita), “Gini coefficient”, “Marriage rate”, “Divorce rate”, “Migration balance”, “Physician density”, “Nurse density”, “Number of hospitals” and demographic indicators (crude birth rate (CBR), crude mortality rate (CMR)) were extracted from national and regional statistical reference books [18]. Given the retrospective nature of our study and its reliance on secondary data, the Ethics Committee of Karaganda Medical University has granted a waiver for informed consent.

Data analysis was conducted using SPSS version 26.0. Skewness was observed in two indicators, namely “The degree of atmospheric pollution” and “GRP per capita”, requiring a normal logarithmic transformation to achieve a more symmetrical distribution for these variables. Correlation analysis was employed to identify relationships among demographic, climatic, environmental, and socioeconomic variables. Subsequently, multidimensional regression models were constructed using regression analysis, and the statistical significance of these models was based on a 95% confidence interval (95% CI).

## 3. Results

Descriptive statistics were used to summarize general trends, variance, degree of distribution normality and homoscedasticity (Table 1).

The mean CBR and CMR were 20.280 and 9.428, respectively, with standard deviations reflecting variability in these demographic indicators. The mean “Degree of atmospheric pollution” across the studied cities was 2.664 ± 1.052, indicating a moderate average pollution level with a notable degree of variability. The ET exhibited a mean of 21.255, indicating a relatively warm climate on average, with a standard deviation of 4.197, suggesting some variability in temperature conditions.

For economic indicators, the mean “GRP per capita” was 971.740, with a substantial standard deviation of 667.658, reflecting notable disparities in economic output across the regions. The “Gini coefficient” mean of 0.260, coupled with a standard deviation of 0.029, indicated a relatively low average income inequality level with limited variability.

Healthcare-related variables, such as “Physician density” and “Nurse density”, displayed means of 36.52 and 97.08 per 10,000 population, respectively, with standard deviations indicating variability in the availability of healthcare professionals.

The correlation coefficients in Table 2 reveal relationships between dependent variables (CBR and CMR) and independent variables.

Strong correlations were identified in Table 2. “Effective temperature” (X2) exhibited a strong positive correlation with CBR (r = 0.924, *p* < 0.01) and a strong negative correlation with CMR (r = −0.802, *p* < 0.01). “Divorce rate” (X6) showed a strong negative correlation with CBR (r = −0.792, *p* < 0.01) and a strong positive correlation with CMR (r = 0.706, *p* < 0.01). The “Gini coefficient” (X4) displayed a strong negative correlation with CBR (r = −0.709, *p* < 0.01) and a notable positive correlation with CMR (r = 0.685, *p* < 0.01). “Physician density” (X8) displayed a weak negative correlation with CBR (r = −0.244, *p* < 0.05) and a weak positive correlation with CMR (r = 0.146, *p* > 0.05). 

Since correlation analysis does not imply a causal relationship, multiple regression analysis was performed to identify the cumulative effect of independent variables on the dependent one. Linear regression analysis was performed with the following predictors: Dependent variable = β0 + β1·X1 + β2·X2 + β3·X3 + β4·X4 + β5·X5 + β6·X6 + β7·X7 + β8·X8 + β9·X9 + β10·X10.

Table 3 presents the results of linear regression analysis for factors associated with the dependent variable CBR. Unstandardized coefficients (β), standard errors (SE), standardized coefficients (β), t-values, significance (Sig.), and variance inflation factors (VIF) were provided for each predictor. The constant term is 8.685 (SE = 8.007), with a t-value of 1.085 and a nonsignificant *p*-value of 0.283. Noteworthy predictors included the positive influence of effective temperature (β = 0.842, t = 7.645, *p* < 0.0001) and marriage rate (β = 0.780, t = 5.674, *p* < 0.0001), while the Gini coefficient (β = −27.342, t = −2.398, *p* = 0.020) and divorce rate (β = −2.060, t = −3.824, *p* < 0.0001) exhibited negative associations. Other variables, such as the ln-transformed degree of atmospheric pollution, ln-transformed GRP per capita, migration balance, physician density, nurse density and number of hospitals, also contributed to the model. The overall model performance was high (R = 0.970, R^2^ = 0.940, R^2^ adjusted = 0.930), with an F-statistic of 90.01 (*p* < 0.0001), indicating a statistically significant relationship between the predictors and the CBR.

The assessment of the assumption of multidimensional model linearity revealed the absence of extreme outliers in our dataset, as indicated by the analysis of residue statistics, where the maximum and minimum values of standard residual values were −2.861 and 2.180, respectively. The Durbin–Watson statistic value of 1.534 allowed us to infer the independence of observations, as it fell within the acceptable range. Figure 2 depicts a graph of the normal probability of standardized model residuals, where the regression line either traversed through the data points or closely approximated them. The scattering diagram confirmed that the standardized remnants of the model were relatively evenly distributed on both sides of 0, signifying the validity of the model and the establishment of homoscedasticity. Low collinearity was evident, as indicated by variance inflation coefficients (VIF) values below the critical threshold (<10).

The assessment of the percentage of explanations for changes in the dependent variable in linear models typically relies on the coefficient of determination. However, given its tendency to inflate with a large number of observations, our focus turned to the values of the adjusted coefficient of determination. The model revealed that the anticipated combined effect of these six predictors, as indicated by the adjusted R-squared (R^2^ adjusted), can account for 93.0% of the changes observed in the CBR.

In the presented linear regression analysis for the CMR (Table 4), the degree of atmospheric pollution (β = −0.949, *p* = 0.044) and the effective temperature (β = −0.294, *p* < 0.0001) could be linked to lower mortality rates. The Gini coefficient showed a positive correlation with mortality (β = 19.923, *p* = 0.015), implying that higher income inequality was associated with increased mortality. Variables such as GRP per capita, marriage rate, divorce rate, migration balance, physician density, nurse density and number of hospitals contribute to the model, although their individual impacts varied. The overall model demonstrated a good fit (R^2^ = 0.796).

The assessment of the percentage of explanations for changes in the dependent variable in linear models typically relies on the coefficient of determination. However, given its tendency to inflate with a large number of observations, our focus turned to the values of the adjusted coefficient of determination. The model revealed that the anticipated combined effect of these six predictors, as indicated by the adjusted R-squared (R^2^ adjusted), can account for 93.0% of the changes observed in the CBR.

Thus, both CBR and CMR regression models demonstrated a significantly high explanatory power. The most substantial prognostic contribution to changes in dependent variables, with intricate and combined effects of factors, was consistently observed in all models through the independent variable ‘Effective temperature’.

## 4. Discussion

Central Asia is poised to undergo faster relative population growth (36.9%) by the mid-21st century compared to the global average (26.0%). The demographic landscape in the region, notably Kazakhstan, reflects a complex interplay of historical, social, economic, and political factors. Kazakhstan has experienced significant demographic shifts in recent decades, with indicators crucial for economic development indicating a favorable demographic profile for the country and the broader Central Asian region over the next three decades [1]. Kazakhstan was chosen as a focal point for several reasons. First, its sharply continental climate, characterized by hot summers, capricious springs and autumns and cold winters with blizzards, adds an element of variability and unpredictability. Second, the nation grapples with substantial air pollution issues, ranking 64th globally in the air pollution index [19]. Third, development indicators, specifically a “prosperity” rating, place Kazakhstan in the category of countries with a moderate level of well-being, predicting a growth rate of 3.6% annually, allowing the country to maintain its GDP per capita relative to the world [20,21].

In our analysis, we sought to understand the influence of climatic, environmental, and socioeconomic factors on population indicators. This required us to distinguish between primary and secondary influences. We identified ten primary factors in our study and utilized correlation analysis to investigate their influence. The results showed an interesting pattern: the correlations with the dependent variables for CBR were diametrically opposed to the correlations with the CMR. This suggests a complex interplay between these different predictors and demographic outcomes. While strong statistical correlations between individual socioeconomic and climatic–ecological variables and demographic variables were not readily apparent, weaker and moderate correlations emerged when considering a common causal effect. These types of connections can sometimes lead to misunderstandings about how different factors relate if the underlying mechanisms are unknown. To explore this further, we constructed three linear regression models using all variables to analyze the relationships with CBR and CMR. These models help illuminate how these diverse factors combine to shape population dynamics. This kind of comprehensive approach is essential for describing the intricate ways that broader social, economic, climatic, and ecological conditions impact demographic trends.

In our study, both “Effective temperature” (β = 0.842, *p* < 0.0001) and “Marriage rate” (β = 0.780, *p* < 0.0001) showed significant positive associations with the crude birth rate. Regions with higher effective temperatures and marriage rates tended to have higher birth rates. The Gini coefficient (β = −27.342, *p* = 0.020) and divorce rate (β = −2.060, *p* < 0.0001) exhibited negative associations with the crude birth rate. Higher income inequality and lower divorce rates were linked to lower birth rates.

The linear regression analysis explored factors influencing the CMR in the context of various predictors. Notable findings included a significant negative association between the “Degree of atmospheric pollution_ln” and the mortality rate (β = −0.949, *p* = 0.044), indicating that higher atmospheric pollution is linked to lower mortality. Conversely, “Effective temperature” exhibited a significant negative influence (β = −0.294, *p* < 0.0001), implying that regions with lower temperatures experience higher mortality rates. The “Gini coefficient” showed a positive correlation with mortality (β = 19.923, *p* = 0.015), suggesting an impact of income inequality on mortality. The model, overall, demonstrated a high fit (R^2^ = 0.796), with a significant F-statistic (F = 22.19, *p* < 0.0001), underscoring the statistical significance of the relationship between the predictors and the crude mortality rate.

Four to six predictors in the regression models of indicators characterizing the processes of population natural movement were most predictively related: “Effective temperature”, “Marriage rate”, “Divorce rate”, “Gini coefficient”, “Physician density, per 10,000 population”, “Nurse density, per 10,000 population” and “Degree of atmospheric pollution_ln”, with *p*-values less than 0.05. “Per capita population_ln”, “Marriage rate” and “Number of hospitals” were not of high importance (*p* > 0.05).

The fact that the climate-related variable («Effective temperature») played the most prominent role as a predictor is consistent with the findings of many other studies. According to various researchers, seasonal changes in solar activity, temperature, humidity and the chemical composition of the air can have a significant impact on the health of the population and demographics [22,23,24].

The “Divorce rate” predictor can also have an impact on demographic processes. Demography considers divorce as one of the factors affecting the formation of the marital structure of the population, which is closely related to the reproduction of the population and is the most important demographic factor in terms of fertility, family formation, and changes in the family structure of the population. In accordance with this, the higher the divorce rate in the republic, the lower the birth rate [25].

The negative values of the standardized coefficient β for the “Gini coefficient” predictor indicates a positive birth rate dynamic when the Gini coefficient decreases, while positive values of β suggest a negative mortality rate dynamic as the Gini coefficient increases. This allows us to assume that the Gini coefficient reflects a set of institutional factors affecting the reproductive behavior of the population as a whole [13].

In general, the results demonstrated that economic indicators themselves were weaker predictors of the natural movement of the population; however, at the same time, GDP clearly mediates the effect of predictor variables on life expectancy and other demographic indicators, although the extent of the impact varies [14,26].

The results showed the presence of the mean correlation between variables related to the number of medical workers and hospital facilities, which indicates that the impact of medical services is rather indirect [27]. In regions characterized by high population densities, it becomes important to maintain an adequate number of healthcare professionals. This is necessary to ensure the effective management of health services, given the potential strain on resources. Conversely, in areas with lower population densities or those classified as rural, it is critical to sustain a sufficient density of physicians and nurses. This provision ensures that necessary medical care remains accessible to all individuals, mitigating the need for extensive travel.

Even though the correlation between the “Degree of atmospheric pollution” predictor and demographic indicators appears to be low, it is crucial not to disregard the synergistic effects when exposed to air pollutants such as suspended particles, nitrogen oxide, and hydrogen sulfide. These elements collectively contribute to a cumulative effect on public health, despite their individual weak correlations with demographic indicators. A considerable body of research confirms that there is a compounded impact of air pollution and weather conditions on both mortality rates and fertility levels [28,29,30]. The combined influence of these factors can result in significant health issues, including increased mortality rates and decreased fertility. This points toward a complex interaction where various environmental factors work together, aggravating their individual impacts. Moreover, it is important to note that these relationships may vary depending on other socioeconomic or geographical factors. Therefore, further research should dig deeper into understanding these complex interactions and their implications for population health.

Several limitations should be acknowledged in this study. First, the geographic scope was limited to ten main cities in Kazakhstan, potentially restricting the generalizability of our findings to other regions or rural settings. Second, we relied on secondary data from official sources, which might have inherent limitations related to quality, completeness and accuracy. Third, the use of “Effective temperature” as a bioclimatic indicator, based primarily on temperature and humidity effects, could limit its applicability under varying climatic conditions. Fourth, skewness observed in two indicators necessitated logarithmic transformations that could have impacted the outcomes. Fifth, the complex link between the multitude of predictors and demographic outcomes may complicate the interpretation of results. Therefore, this warrants further investigation. Finally, there are likely other unmeasured variables like cultural norms or policy changes that may also impact demographic trends. These limitations suggest caution when generalizing these findings beyond their specific context.

## 5. Conclusions

This study reveals findings pertaining to the significant associations between demographic indicators, namely the crude birth rate and mortality rate, and various climatic, environmental, and socioeconomic factors in Kazakhstan. The effective temperature was shown to have a notable positive correlation with the birth rate, while marriage rates also showed a substantial positive effect. Conversely, factors such as income inequality (as indicated by the Gini coefficient) and divorce rates were found to negatively impact the crude birth rate. Our research confirms that environmental conditions and socioeconomic status significantly influence population health demographics. While economic indicators were relatively weaker predictors of population dynamics, GDP was identified as having a mediating role on life expectancy and other demographic indicators. Medical services seemed to exert an indirect influence on demographic processes. The results obtained underscore the importance of climate-related variables like effective temperature on population health demographics. Our analysis emphasizes that understanding these connections between various factors can aid in implementing specific measures aimed at addressing public health issues and improving the quality of life in Kazakhstan. This is particularly relevant given the expected rapid population growth in Central Asia this century.

## Figures and Tables

**Figure 1 ijerph-21-00416-f001:**
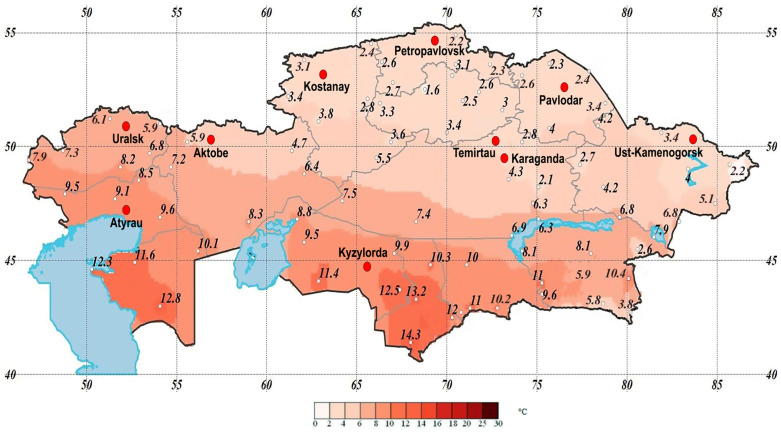
Average annual temperature in Kazakhstan, °C.

**Figure 2 ijerph-21-00416-f002:**
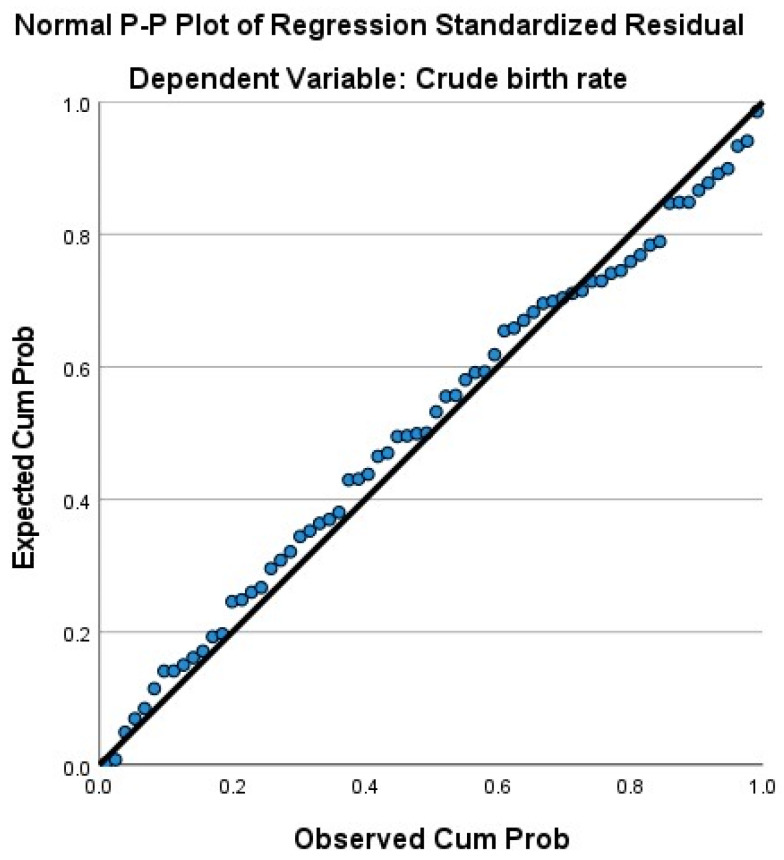
Normal R-R regression graph of standardized residuals.

**Table 1 ijerph-21-00416-t001:** Descriptive statistics of dependent and independent variables.

Variables	Mean	Median	Std. Deviation	Skewness	Kurtosis
Degree of atmospheric pollution	2.664	2.300	1.052	1.333	1.977
Effective temperature	21.255	19.500	4.197	0.898	−0.570
GRP per capita	971.740	676.250	667.658	1.613	1.503
Gini coefficient	0.260	0.260	0.029	−0.521	−0.730
Marriage rate	9.909	9.805	1.496	0.171	−0.608
Divorce rate	4.122	4.365	0.736	−0.523	−0.922
Migration balance	1933.659	1793.500	1268.137	0.579	−0.505
Physician density, per 10,000	36.516	37.900	7.647	−0.069	−1.406
Nurse density, per 10,000	97.077	99.900	13.442	−0.312	−0.387
Number of hospitals	53.417	46.000	25.231	0.770	−0.690
* Crude birth rate	20.280	17.010	5.701	0.657	−1.109
* Crude mortality rate	9.428	9.885	2.127	−0.260	−0.916

Note: * Dependent variables.

**Table 2 ijerph-21-00416-t002:** Correlation coefficients between CBR, CMR and independent variables.

Code	Independent Variable	CBR	CMR
		Y1	Y2
X1	Degree of atmospheric pollution_ln	−0.088	−0.066
X2	Effective temperature	0.924 **	−0.802 **
X3	GRP per capita_ln	0.052	−0.106
X4	Gini coefficient	−0.709 **	0.685 **
X5	Marriage rate	0.262 *	−0.081
X6	Divorce rate	−0.792 **	0.706 **
X7	Migration balance	0.312 **	−0.418 **
X8	Physician density, per 10,000	−0.244 *	0.146
X9	Nurse density, per 10,000	−0.056	0.203
X10	Number of hospitals	−0.389 **	0.252 *

* Correlation is significant at 0.05 (two-way). ** Correlation is significant at 0.01 (two-way).

**Table 3 ijerph-21-00416-t003:** Linear regression analysis of factors influencing the ‘Crude birth rate’ dependent variable.

Variables	Unstandardized Coefficients		Standardized Coefficients	t	Sig.	VIF
	β	SE	β			
(Constant)	8.685	8.007		1.085	0.283	
Degree of atmospheric pollution_ln	1.200	0.658	0.071	1.823	0.074	1.468
Effective temperature	0.842	0.110	0.645	7.645	0.000	6.805
GRP per capita_ln	−0.852	0.544	−0.076	−1.567	0.123	2.269
Gini coefficient	−27.342	11.402	−0.139	−2.398	0.020	3.233
Marriage rate	0.780	0.137	0.208	5.674	0.000	1.282
Divorce rate	−2.060	0.539	−0.261	−3.824	0.000	4.447
Migration balance	0.000	0.000	0.034	0.931	0.356	1.293
Physician density, per 10,000	0.083	0.037	0.106	2.223	0.030	2.186
Nurse density, per 10,000	0.037	0.018	0.092	2.116	0.039	1.808
Number of hospitals	−0.005	0.012	−0.021	−0.459	0.648	2.004

Dependent variable: Crude birth rate; R = 0.970; R^2^ = 0.940; R^2^ adjusted = 0.930; F (10.57) = 90.01 (*p* < 0.0001).

**Table 4 ijerph-21-00416-t004:** Linear regression analysis of factors influencing the ‘Crude mortality rate’ dependent variable.

Variables	Unstandardized Coefficients		Standardized Coefficients	t	Sig.	VIF
	β	SE	β			
(Constant)	7.194	5.601		1.284	0.204	
Degree of atmospheric pollution_ln	−0.949	0.460	−0.150	−2.061	0.044	1.468
Effective temperature	−0.294	0.077	−0.595	−3.812	0.000	6.805
GRP per capita_ln	0.601	0.380	0.143	1.581	0.119	2.269
Gini coefficient	19.923	7.976	0.269	2.498	0.015	3.233
Marriage rate	0.036	0.096	0.025	0.373	0.710	1.282
Divorce rate	0.404	0.377	0.135	1.071	0.289	4.447
Migration balance	0.0001	0.000	−0.193	−2.840	0.006	1.293
Physician density, per 10,000	−0.044	0.026	−0.150	−1.695	0.095	2.186
Nurse density, per 10,000	0.006	0.012	0.039	0.490	0.626	1.808
Number of hospitals	−0.006	0.008	−0.058	−0.690	0.493	2.004

Dependent variable: Crude mortality rate; R = 0.892; R^2^ = 0.796; R^2^ adjusted = 0.760; F (10.57) = 22.19 (*p* < 0.0001).

## Data Availability

Data were derived from public domain resources: Agency for Strategic planning and reforms of the Republic of Kazakhstan Bureau of National statistics—Main. Available online: https://stat.gov.kz/en/ (accessed on 6 December 2023).

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
