# Peer review of "The Role of Climatic, Environmental and Socioeconomic Factors in the Natural Movement of Urban Populations in Kazakhstan, 2012–2020: An Analysis from a Middle-Income Country in Central Asia"

_ijerph, 2024, doi:10.3390/ijerph21040416_

Round 1

Reviewer 1 Report

Comments and Suggestions for Authors

Comments to the Author

The topic on
The Role of Climatic, Environmental and Socioeconomic Factors in the Natural Movement of Urban Population in Kazakhstan, 2012-2020: an Analysis from a Middle-Income Country in Central Asia is interesting. Regarding this study, there are some comments and critical questions that should be answered before publishing the paper.

a. This study aims
to conduct a complex assessment, evaluating the influence of climatological, environmental, and socioeconomic factors on the dynamics of natural population movement in the Republic of Kazakhstan.

I think you need to explain about a complex asssement in your research context. In methodology parts, it seems ecological studies. The trend and time series analysis is suggeted

b.  Correlation analysis was employed to identify relationships among demographic, climatic, environmental, and socioeconomic variables. Subsequently, multidimensional regression models were constructed using regression analysis, and the statistical significance of these models was based on a 95% confidence interval (95% CI)

Sorry, I am not familiar with the multidimensional regression model. May you explain the basic principle of that methods?

Other comments

Line 64

Information was collected from 10 main cities…………………..

Comment: Add the map of Kazakhstan is suggested. In addition, displaying a change of climatic, environmental and socioeconomic  variables in Kazakhstan  over the period 2012-2024 may be interesting to explore the spatial distribution and trends of natural movement of urban population

Line 70-71

Weather conditions were assessed using the "Effective temperature" (ET) bioclimatic indicator, a composite measure incorporating temperature and humidity effects

Comment:

I am curious why you use the term Effective temperature instead of Average Temperature and Humidity. What is the unit of Effective Temperature? (you can elaborate this  on the discussion part)

Lines 91-92

Descriptive statistics were used to summarize general trends, variance, degree of distribution normality, and homoscedasticity (Table 1).

Comment:

I find it difficult to understand to looking at your table. I suggest you r revise it accordingly

Line 154

Figure 1. Normal R-R regression graph of standardized residuals

Comment:

What is the importance to show this graph to readers?

Line 243

The stability of the “Gini coefficient” effect

Comment:

 May you explain the role of stability on the Gini coefficient on your study

Author Response

Dear Reviewer,

Thank you very much for your insightful comments and suggestions. Here are our responses:

Comment 1 “I think you need to explain about a complex asssement in your research context. In methodology parts, it seems ecological studies. The trend and time series analysis is suggeted”:

Response to comment 1: Complex assessment in our study refers to the comprehensive examination of various climatological, environmental, and socioeconomic factors that influence population movement. We agree with your suggestion about the ecological approach and time series analysis, which we will consider incorporating in future research.

Comment 2 “Sorry, I am not familiar with the multidimensional regression model. May you explain the basic principle of that methods?”:

Response to comment 2: Multidimensional regression models are an extension of the simple linear regression model where multiple independent variables are used to predict a dependent variable. Our model allows us to assess how each predictor variable (climatic, environmental, socioeconomic) contributes to changes in population natural movement indicators while controlling for other variables.

Comment 3: “Add the map of Kazakhstan is suggested. In addition, displaying a change of climatic, environmental and socioeconomic  variables in Kazakhstan  over the period 2012-2024 may be interesting to explore the spatial distribution and trends of natural movement of urban population

Response to comment 3: "We appreciate the suggestion. In the methods section, we have added a map of Kazakhstan showing the annual average temperature distribution across the entire territory. Unfortunately, at present, we are unable to provide a visual spatial distribution of other parameters on the map."

Figure 1. Average annual temperature in Kazakhstan, °C.

Comment 4: “I am curious why you use the term Effective temperature instead of Average Temperature and Humidity. What is the unit of Effective Temperature? (you can elaborate this  on the discussion part)

Response to comment 4: "Effective temperature" is a bioclimatic indicator that integrates both temperature and humidity effects into a single measure that can better represent human comfort level in varying weather conditions than average temperature or humidity alone. Its units are degrees Celsius (°C). We added this description to the Methods part. Various applications of this indicator can be found on https://www.sciencedirect.com/topics/engineering/effective-temperature

Comment 5: “I find it difficult to understand to looking at your table. I suggest you revise it accordingly”

Response to comment 5: We apologize if our table was unclear. In the revision, we simplified the title to make it more understandable and marked the dependent variables with  *.

Comment 5: “Line 154 Figure 1. Normal R-R regression graph of standardized residuals

What is the importance to show this graph to readers?”

Response to comment 5: This graph illustrates the homoscedasticity of residuals — an essential assumption in regression analysis — assuring readers about the appropriateness of our statistical model. Nonetheless, we understand its relevance may not be immediately apparent, however believe us it is important to test it and show using regression models.

We can also see it on the residuals’ distribution histogram

 and on the scatterplot

Comment 6: Line 243 The stability of the “Gini coefficient” effect. May you explain the role of stability on the Gini coefficient on your study

Response to comment 6: We appreciate your feedback on the unclarity of our statement related to the Gini coefficient. In response, we have rewritten this paragraph to provide a more precise and understandable explanation. Currently, the statement looks like this in the discussion part:

“The negative values of the Standardized Coefficients β for the "Gini coefficient" predictor indicate a positive birth rate dynamic when the Gini coefficient decreases, positive values of β suggest a negative mortality rate dynamic as the Gini coefficient increases. This allows us to assume that the Gini coefficient reflects a set of institution-al factors affecting the reproductive behavior of the population as a whole.”

We would like to express our sincere thanks to the reviewer for such thorough and detailed feedback. Your insights have been critical in refining our manuscript, allowing us to improve it. We greatly appreciate the time and effort you've taken in reviewing our work.

Reviewer 2 Report

Comments and Suggestions for Authors

The paper clearly states the objectives, the hypothesis, and methods.  I have no additional suggestions fort the authors.

Comments on the Quality of English Language

The English is very good but as a suggestion some words could be more colloquial…. Professional but colloquial.  For example in line 33-35 …. The demographic projections of the United Nations for 2019 indicate (states) that the world's demographic landscape in the 21st century will be shaped by ongoing trends, encompassing (about) a decline in fertility rates, an increase in life expectancy, demographic transitions, migration, and other factors…

Author Response

Dear Reviewer,

Thank you so much for your constructive feedback and valuable suggestions. We have carefully reviewed them and made the necessary amendments in our manuscript, particularly focusing on adopting a more colloquial yet professional tone where it is possible as advised:

In line 33 we have implemented your suggestion for particular wording “…The demographic projections of the United Nations for 2019 indicate state that the world's…” and instead of 2019 we were supposed to write 2022.

In line 34 instead of “…demographic landscape in the 21st century will be shaped by ongoing trends,encompassing including…”

In lines 36-38 instead of whole sentence “Fundamental to the health status of populations are demographic indicators, with acknowledgment that climatic conditions, anthropogenic environmental pollution, and the lifestyle of the populace play pivotal roles.”

We wrote “Demographic indicators are crucial to the health status of populations, acknowledging that climatic conditions, human-induced environmental pollution, and the lifestyle choices of the population play significant roles.”

In lines 38-44: instead of “The industrial nature of many regions in Kazakhstan, featuring mining, metallurgical, chemical, and energy enterprises, presents a significant ecological challenge, impacting both the region's ecology and public health. Thus, this study categorizes factors into three groups: climatic, environmental, and socioeconomic, emphasizing the need for systemic analysis methods, estimates, and predictive models to comprehend the contribution of environmental phenomena to demographic processes.

We splitted the statements into several sentences to make it more understandable for readers “Kazakhstan presents an interesting case study due to its industrial nature. Dominated by mining, metallurgical, chemical, and energy enterprises, these aspects present notable ecological challenges that impact not only the local ecology but also public health. Recognizing this confluence of factors, our study divides them into three broad categories: climatic, environmental, and socioeconomic. It emphasizes the need for comprehensive analysis methods, estimates, and predictive models to understand how these environmental phenomena affect population dynamics”.

In lines 51-53: We improved the statement from “The healthcare system's effectiveness emerges as a key factor in increasing life expectancy, and some studies suggest a correlation between a high gross domestic product (GDP) per capita and greater life expectancy”: to “Meanwhile, healthcare system efficiency plays an important role in enhancing life expectancy; several studies even suggest a correlation between higher GDP per capita and increased life expectancy” and extended the description of the healthcare-related factors.

In lines 253-257 we tried to explain some points with plain language and extended our last statements of the whole manuscript.

Currently, this part looks like this “Even though the correlation between the "Degree of atmospheric pollution" predictor and demographic indicators appears to be low, it is crucial not to disregard the synergistic effects when exposed to air pollutants such as suspended particles, nitrogen oxide, and hydrogen sulfide. These elements collectively contribute to a cumulative ef-fect on public health, despite their individual weak correlations with demographic indicators. A considerable body of research confirms that there is a compounded impact of air pollution and weather conditions on both mortality rates and fertility levels [26, 27,28]. The combined influence of these factors can result in significant health issues, including increased mortality rates and decreased fertility. This points towards a complex interaction where various environmental factors work together, aggravating their individual impacts. Moreover, it's important to note that these relationships may vary depending on other socioeconomic or geographical factors. Therefore, further research should dig deeper into understanding these complex interactions and their implica-tions for population health.”

We appreciate your positive remarks about our work and also your help in identifying areas that needed improvement. In response to your insights, we have also revised and updated our conclusions to better reflect the objectives of our research:

Currently, the conclusions part looks like this:

This study reveals findings pertaining to the significant associations between demographic indicators, namely the crude birth rate and mortality rate, and various climatic, environmental, and socioeconomic factors in Kazakhstan. The effective temperature was shown to have a notable positive correlation with the birth rate, while marriage rates also showed a substantial positive effect. Conversely, factors such as income inequality (as indicated by the Gini coefficient) and divorce rates were found to negatively impact the crude birth rate. Our research confirms that environmental conditions and socioeconomic status significantly influence population health demographics. While economic indicators were relatively weaker predictors of population dynamics, GDP was identified as having a mediating role on life expectancy and other demographic indicators. Medical services seemed to exert an indirect influence on demographic processes. The results obtained underscore the importance of climate-related variables like effective temperature on population health demographics. Our analysis emphasizes that understanding these connections between various factors can aid in implementing specific measures aimed at addressing public health issues and improving quality of life in Kazakhstan. This is particularly relevant given the expected rapid population growth in Central Asia this century.

Thank you again for your time and thoughtful input.

Reviewer 3 Report

Comments and Suggestions for Authors

The paper was written very clearly and the objectives were established.

Methods: Authors should state if they received ethical approval for the study (or if it was exempted).

Discussion: State any limitations of the study which the authors had to consider or grapple with. Was the use of secondary data an issue?

Is there any explanation which the authors can put forward for findings on the 'degree of atmospheric pollution'?

Conclusions:

Authors would be advised to note that further research to confirm the findings of the study may be warranted - especially in light of counter-intuitive findings such as the 'degree of atmospheric pressure' and negative correlations between the number of hospitals and CBR.

Table 4 title - should read Crude Mortality Rate - CMR 

Author Response

Dear Reviewer,

Thank you so much for your constructive feedback. We truly appreciate your thoughtful comments and valuable feedback on our paper. We have carefully reviewed them and made the necessary amendments in our manuscript:

  1. We agree with you that we have to respect the Ethics reporting protocols. We added the following statement “Given the retrospective nature of our study and its reliance on secondary data, the Ethics Committee of Karaganda Medical University has granted a waiver for informed consent” to the methods part.
  2. As for limitations, indeed, one of the challenges we faced was the use of secondary data. Despite its broad reach, it may not accurately capture all relevant variables nor allow for control over data quality or completeness. We acknowledge this as a potential limitation in our research, as well as other points that we added as a final paragraph to our discussion section:

“Several limitations should be acknowledged in this study. Firstly, the geographic scope was limited to ten main cities in Kazakhstan, potentially restricting the generalizability of our findings to other regions or rural settings. Second, we relied on secondary data from official sources which might have inherent limitations related to quality, completeness, and accuracy. Third, the use of "Effective temperature" as a bioclimatic indicator, based primarily on temperature and humidity effects, could limit its applicability under varying climatic conditions. Fourthly, skewness observed in two indicators required logarithmic transformations that could have impacted the outcomes. Fifthly, the complex link between the multitude of predictors and demographic outcomes may complicate interpretation of results. Therefore, this warrant further investigation. Finally, there are likely other unmeasured variables like cultural norms or policy changes that may also impact demographic trends. These limitations suggest caution when generalizing these findings beyond their specific context.”

  1. Concerning findings on the 'degree of atmospheric pollution', we propose that these are likely due to complex interactions between various factors affecting population health beyond just pollution levels. These could include socioeconomic status, access to healthcare, and other environmental conditions.
  2. We agree with your suggestion and recognize that further research is indeed necessary to confirm our findings. We added these points to our limitations part of the discussion.
  3. Finally, thank you for pointing out the error in Table 4's title - we corrected it : “Table 4. Linear regression analysis of factors influencing the 'Crude mortality rate' dependent variable.”

We are very grateful for your time in reviewing our work and providing insightful suggestions for improvement.